# Impact of iron fortification on anaemia and iron deficiency among pre-school children living in Rural Ghana

**Samuel Kofi Tchum**[1,2]*, **Fareed Kow Arthur**[1], **Bright Adu**[3], **Samuel Asamoah Sakyi**[4], **Latifatu Alhassan Abubakar**[2], **Dorcas Atibilla**[2], **Seeba Amenga-Etego**[2], **Felix Boakye Oppong**[2], **Francis Dzabeng**[2], **Benjamin Amoani**[5], **Thomas Gyan**[2], **Emmanuel Arhin**[6], **Kwaku Poku-Asante**[2]

**1** Department of Biochemistry and Biotechnology, College of Sciences, Kwame Nkrumah University of Science and Technology, Kumasi, Ghana, **2** Kintampo Health Research Centre, Ghana Health Service, Kintampo-North, Ghana, **3** Department of Immunology, College of Health Sciences, Noguchi Memorial Institute for Medical Research, University of Ghana, Legon, Accra, Ghana, **4** Department of Molecular Medicine, School of Medical Sciences, Kwame Nkrumah University of Science and Technology, Kumasi, Ghana, **5** Department of Biomedical Sciences, School of Allied Health Sciences, University of Cape Coast, Cape Coast, Ghana, **6** Faculty of Earth and Environmental Sciences, Department of Earth Science, C K. Tedam University of Technology and Applied Sciences, Navrongo, Ghana

* kofi.tchum@kintampo-hrc.org

**Data Availability Statement:** Data are available from figshare: https://figshare.com/s/48673d5ea8e5bb2d7c83

**Funding:** The author(s) received no specific funding for this work.

## Abstract

Anaemia in young sub-Saharan African children may be due to the double burden of malaria and iron deficiency. Primary analysis of a double-blind, cluster randomized trial of iron containing micronutrient powder supplementation in Ghanaian children aged 6 to 35 months found no difference in malaria risk between intervention and placebo groups. Here, we performed a secondary analysis of the trial data to assess the impact of long-term prophylactic iron fortificant on the risk of iron deficiency and anaemia in trial subjects. This population-based randomized-cluster trial involved 1958 children aged between 6 to 35 months, identified at home and able to eat semi-solid foods. The intervention group (n = 967) received a daily dose containing 12.5 mg elemental iron (as ferrous fumarate), vitamin A (400 µg), ascorbic acid (30 mg) and zinc (5 mg). The placebo group (n = 991) received a similar micronutrient powder but without iron. Micronutrient powder was provided daily to both groups for 5 months. At baseline and endline, health assessment questionnaires were administered and blood samples collected for analysis. The two groups had similar baseline anthropometry, anaemia, iron status, demographic characteristics, and dietary intakes (p > 0.05). Of the 1904 (97.2%) children who remained at the end of the intervention, the intervention group had significantly higher haemoglobin (p = 0.0001) and serum ferritin (p = 0.0002) levels than the placebo group. Soluble transferrin receptor levels were more saturated among children from the iron group compared to non-iron group (p = 0.012). Anaemia status in the iron group improved compared to the placebo group (p = 0.03). Continued long-term routine use of micronutrient powder containing prophylactic iron reduced anaemia, iron deficiency and iron deficiency anaemia among pre-school children living in rural Ghana's malaria endemic area.

**Competing interests:** The authors have declared that no competing interests exist.

## Introduction

Anaemia prevalence in children under 5 years old was estimated to be 43% globally in 2011 and was higher (71%) in central and western Africa [1]. In Ghana, the prevalence of anaemia, iron deficiency and iron deficiency anaemia among pre-school children in 2017 were 35.6%, 21.5% and 12.2% respectively [2]. Early childhood anaemia has been associated with reduced cognitive ability, developmental delays and disability [3,4]. Similarly, zinc deficiency is thought to be as prevalent as iron deficiency affecting about 293 million children below the age of five years [5]. Despite the significant physiological roles of micronutrient in human health, their deficiencies are a universal health burden, particularly among young children in developing countries [6]. The multiple micronutrient powder (MNP) contains a mixture of at least iron, zinc, and vitamin A and is recommended by the World Health Organization (WHO) as a treatment therapy to prevent malnutrition in children and during health emergencies [7]. As an intervention to curb the universal health burden of micronutrient deficiencies, vitamin A and zinc supplementation for children, and fortification of foods with iron and iodine, have been considered the most cost-effective approach [8]. MNP with iron (iron-MNP) given to pre-school children improved their motor and cognitive performance and mitigated severe anaemia [9,10].

In malaria endemic regions, morbidity due to iron deficiency anaemia may be further worsened by *Plasmodium falciparum* infections characterized by excessive breakdown of both infected and uninfected erythrocytes thus contributing to lower blood haemoglobin levels [11–15]. More data is needed to ascertain whether provision of long-term continued prophylactic iron-MNP to children aged 6 to 35 months living in malaria endemic areas will reduce iron deficiency anaemia (IDA) [16–18]. In 2003, a randomized placebo-controlled trial conducted in Pemba, Zanzibar involving 32,000 pre-school Tanzanian children was stopped promptly on the advice from the trial's Data Safety Monitoring Board (DSMD) due to higher hospitalizations or mortality rate in the iron group [19]. However, a further secondary subgroup analysis involving recruited iron-replete children at baseline studies discovered a limitation on the risk of adverse events resulting in ethical difficulties and complicated study designs in malaria endemic areas [19]. The UNICEF and WHO joint statement was uncertain about MNP use, since the absorption characteristics differ considerably from iron syrups or tablets if given to children aged between 6 and 36 months [20]. In 2006, the joint statement was amended to specifically recommend home fortification of foods plus iron-MNP to children at risk of iron deficiency and anaemia [21]. Indeed, primary analysis of the current trial data showed no difference in malaria risk between the iron-MNP and placebo groups [22,23]. The WHO thus recommended in 2016 that in heavily malaria transmission areas, pre-school children at risk of iron deficiency and anaemia should be provided with oral iron intervention if they have access to anti-malaria intervention strategies (insecticide-treated bed-nets, antimalarial drug therapy and vector-control programmes) [24]. Here, we present a secondary analysis of the double-blind, cluster randomized trial data [25], and successfully show the impact of long-term prophylactic iron-MNP on the risk of iron deficiency and anaemia among pre-school children living in malaria endemic areas in Ghana.

## Materials and methods

### Ethical considerations

The study protocol was reviewed and approved by the ethics committees of the Ghana Health Service, Food and Drugs Authority of Ghana, Kintampo Health Research Centre and Hospital for Sick Children (SickKids), Canada. This current study is a secondary analysis of data from the trial registered at ClinicalTrials.gov (Identifier: NCT01001871) [25]. The trial was overseen

by an independent Data and Safety Monitoring Board (DSMB), which was constituted in October 2009 and held three meetings during the trial. Members of the DSMB included international and local health policy makers with expertise in randomized controlled trials, nutrition, paediatrics, statistics and social sciences. The DSMB's statistician summarized the compiled outcome data at the end of the recruitment phase and half-way via the intervention stage for any serious adverse effects. The children's primary caregivers consented to participate in the study. For the interim analysis, if there were any serious adverse events (i.e. hospital admissions or deaths) in the iron group than the non-iron group, the agreement *a priori* was that the study would be terminated.

## Study area and design

This study was a population-based randomized-cluster trial conducted in the Bono (formerly Brong-Ahafo) Region of Ghana. Details of the study area and design have been previously reported. In brief, the study was conducted in Wenchi and Tain, which are two contiguous districts with high malaria transmission in the Region [26]. In 2014, severe-moderate anaemia prevalence among children aged 6–59 months in Bono was 34% according to the Ghana Demographic and Health Survey [1,27]. The trial participants were infants and young children, identified at home and able to eat semi-solid foods (with or without breast milk). For five months, all participants received daily MNP without or with iron (12.5 mg) added to complementary meals. Children who had severe anaemia (haemoglobin < 70.0 g / L), severe malnutrition (weight-for-length z-score < -3.0), or who received iron in supplements within the past 6 months or chronic disease were excluded from the study. The study was conducted in the rainy season during high malaria transmission which allowed for optimum anaemia and iron status assessment.

## Recruitment of subjects

A detailed description of subject recruitment and randomization procedures for the trial has been previously reported elsewhere [25]. Briefly, between March and April 2010, eligible children (6 to 35 months old) were enrolled and randomized (ratio 1:1) to receive either iron-MNP group or MNP with no iron group (MNP-Sprinkles® Mumbai, India). A cluster comprised one or more households in the same compound with at least one child enrolled and to prevent cross contamination between the groups via food sharing, a cluster randomization design was employed using (in house written program) a computer-generated model [25]. This simple random allocation was carried out at the cluster level because the participants lived in same compound with different or extended families in the rural communities. Therefore, this randomization scheme also facilitated the monitoring of participants, particularly those who moved within or outside the study area. All enrolled children received insecticide treated net (ITN). Sachets without or with iron were similar except a subtle 'A' or 'B' labelled markings and both caregivers and study team were blinded. The iron-MNP dose contained elemental iron (12.5 mg) in microencapsulated ferrous fumarate, vitamin A (400 μg), ascorbic acid (30 mg) and zinc (5 mg) [25,28] The control (non-iron) group received a similar MNP without iron. Each day for a 5-month period, 1 sachet of MNP (equivalent of 12.5 mg iron for the iron group) was mixed with a small quantity of semi-solid meals and given to the children. Once a week, field research assistants visited the home of each child to collect morbidity data (including axillary temperature), assess adherence to ITN use and MNP intervention and to supply replenishment. Caregivers saved empty and any unused packets which were counted during visits. The adherence was determined by multiplying the actual number of packages used by 100, divided by the estimated amount used (given perfect adherence). Children were further monitored an extra month without the MNP at the end of the intervention.

## Specimen and data collection

At baseline (BL) and endline (EL) finger or heel prick blood (about 500 μL) sample was collected for haematological, malaria microscopy, acute protein phase and iron biomarker analyses. Blood haemoglobin (Hb) level was measured directly using HemoCue Hb 201$^+$ analyzer (HemoCue AB, Angellholm, Sweden) and severely anaemic were referred for treatment. At baseline, malaria rapid diagnostic test (RDT) (Paracheck *Pf* Ⓡ Device, Orchid Biomedical Systems, Verna, Goa, India) was initially performed and children with positive results were treated for malaria as described [25]. After recovery, participants were enrolled if all other inclusion criteria were met. If a child is febrile (i.e. axillary temperature > 37.5 $^0$C) or febrile 48 hours ago, 100 μL capillary blood sample was collected into 0.5 mL EDTA tube for full blood count, malaria rapid and blood smear test (for parasitaemia and speciation) during the study as previously described [25].

## Specimen processing and analysis

Thin blood films were fixed with methanol and the smears stained with Giemsa for microscopy examination. Each sample slide was read by two independent microscopists and if discrepancy between the two readers was over 50%, a third microscopist was consulted [25]. Full blood count (FBC) was measured by the haematology auto-analyzer (Horiba ABX Micros 60-OT-CT-OS-CS, Montpellier, France) and C-reactive protein (CRP) by immune-turbidi-metrically using the QuikRead 101 analyzer (Orion Diagnostica, Espoo, Finland). Red blood cell zinc protoporphyrin (ZPP) was measured using a haematofluormeter (Model 206D, Aviv Biomedical Inc. Lakewood, NJ, USA). Indirect enzyme-linked immunosorbent assay (ELISA) was used to measure ferritin (Fn) (Spectro Ferritin S-22, Ramco Laboratories Inc. USA) and transferrin receptor (TfR) (TFC-94, Ramco Laboratories Inc. USA) levels as described in the following procedures [25].

## Statistical analysis

We tested the hypothesis that anaemia and iron status would be significantly improved among the children in the iron group than those in the non-iron. Descriptive statistics were used to summarize the study variables. Difference in baseline characteristics between the children in the iron and non-iron group was studied using two-sample Wilcoxon rank-sum (continuous variables) and Chi-squared test for categorical variables.

In studying the effect of the iron intervention on various indicators (anaemia (Hb < 100 g / L), iron deficiency (Ferritin < 30 μg / L, ZPP > 52 μmol / mol haeme), iron deficiency anaemia (IDA), (Hb < 100 g / L, Ferritin < 30 μg / L) status, mixed effect models were used to perform a difference-in-difference analysis while controlling for the compound and individual level correlation [29–31].

Using logistic regression, the risk of anaemia, iron deficiency and iron deficient anaemia were compared between the iron and non-iron group. Generalized estimating equation with robust standard errors was used to obtain population-averaged estimates and to account for the household level clustering. Parameter estimates were reported as odds ratio with 95% confidence intervals. Separate models were considered for anaemia, iron deficiency and iron deficient anaemia. In all the models, we adjusted for child's age (≤ 12 months, 13–24 months and > 24 months) and sex. Acknowledging that ferritin and ZZP interpretation will be confounded by acute phase response, we excluded these indicators for those children who had an elevated CRP (> 8 mg / L) [32,33]. Also, we adjusted for baseline anaemia, baseline iron deficiency and baseline iron deficient anaemia in the model for anaemia, iron deficiency and iron deficient anaemia respectively. All analyses were carried out on an intention-to-treat (ITT)

basis. Data was analysed using STATA (Stata Statistical Software: Release 11. College Station, TX: StataCorp LP, 2015). P < 0.05 was considered statistically significant.

## Results

### Study participant characteristics

The characteristics of the study population have previously been described elsewhere [25]. In summary, a total of 2220 children aged 6–35 months from 22 communities were screened for eligibility and 262 (11.8%) were excluded according to pre-specified criteria. The remaining 1958 children were randomized to receive either prophylactic micronutrient powder with iron (n = 967) or placebo (n = 991). Out of the 1958 children enrolled, blood sample was obtained from 1806 (92.2%) after the MNP intervention Fig 1). By the end of the study (24 weeks), about 3.0% of the participants were lost to follow-up (iron = 67 versus non-iron = 76) for

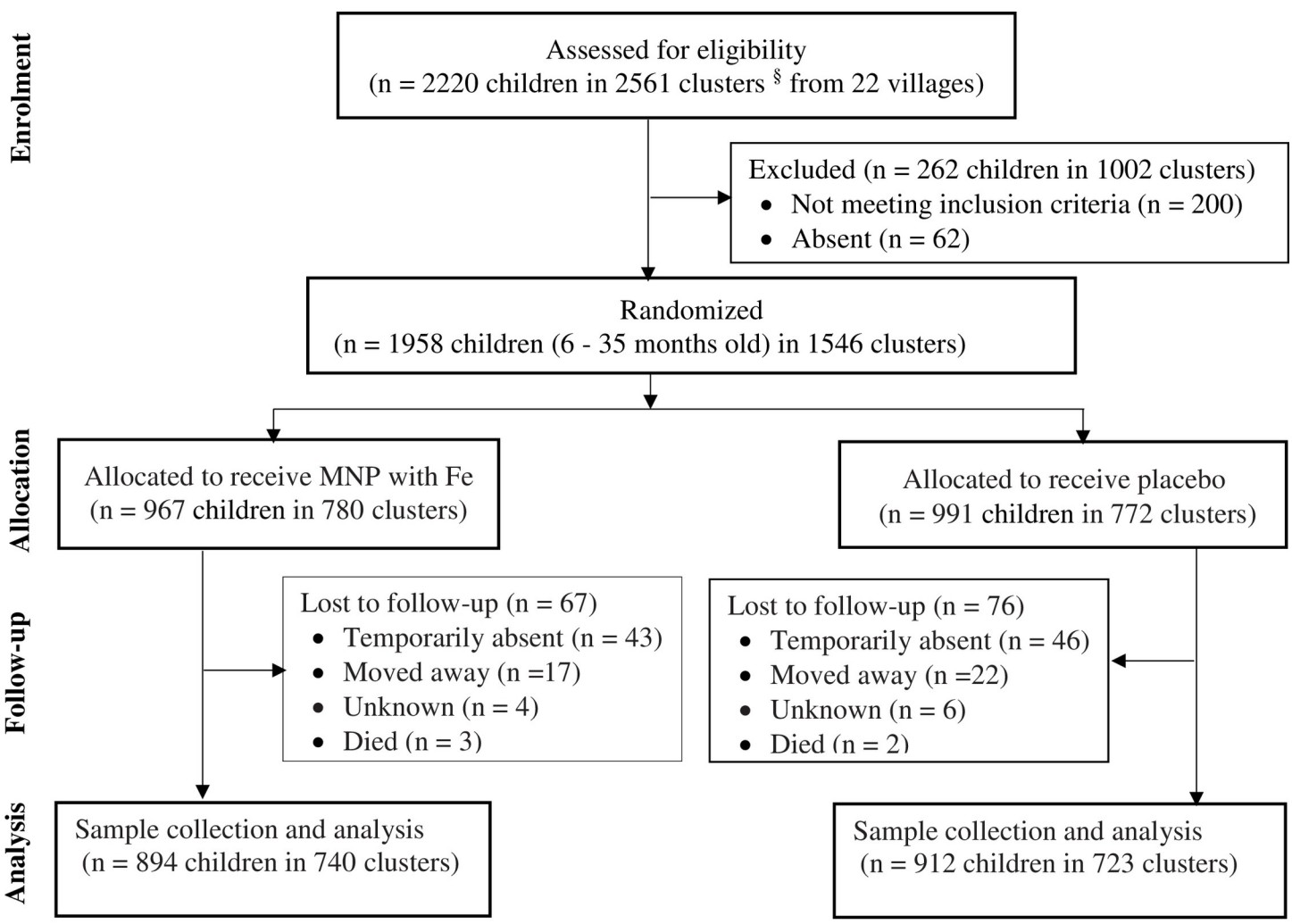

§: A cluster is a compound representing 1 or more households (or family units) living in the same residence with at least 1 eligible child enrolled in the study, MNP: Micronutrient powder, Fe: iron.

**Fig 1. Study profile.** Micronutrient powder (MNP), Fe (iron), Children (CHN), Month (mo), Number (n). §: A cluster is a compound representing 1 or more households (or family units) living in the same residence with at least 1 eligible child enrolled in the study.

863.8 child years of total observation time. The lost to follow-up in both groups was due to families moving from the trial area. Adherence to MNP intervention was similar between the groups and ITN use also did not differ [25]. Baseline characteristics of the children were similar between the groups (Table 1).

## Association of Hb levels between iron and non-iron MNP participants

A mean Hb levels at baseline were comparable between the non-iron and iron groups (10.29 g / dL versus 10.29 g / dL, p = 0.904) (Table 4). However, the overall Hb level in both groups was significantly lower at endline compared to the baseline (p < 0.001). The mean Hb level after the intervention was significantly lower in the non-iron group compared to the iron group (9.31 ± 1.5 g / L versus 9.65 ± 1.7 g / L respectively, p < 0.001). With a difference in difference estimate of 0.35 g / dL (95% CI: 0.19 g / dL—0.51 g / dL, p < 0.001), children in the non-iron group had a mean Hb reduction of 0.35 g / dL compared to those in the iron group (Table 4).

**Table 1. Baseline characteristics amongst the children.**

| Characteristics | Iron Group (n = 894)[a] | Non-iron Group (n = 912)[a] |
|---|---|---|
| **Number of clusters** | 736 | 721 |
| **Cluster size, mean (range)** | 1.6 (1–4) | 1.5 (1–5) |
| **Age, mean (SD), mo** | 19.3 (8.7) | 19.1 (8.6) |
| **Gender, n (%)** | | |
| Male | 458 (51.2) | 457 (50.1) |
| Female | 436 (48.8) | 455 (49.9) |
| **Anthropometric status** | | |
| Wasting, n (%) [95% CI] | 83 (9.3) [7.3–11.0] | 65 (7.1) [5.7–9.1] |
| Weight for length Z-score mean (SD)[b] | -0.54 (3.5) | -0.50 (3.4) |
| Stunted growth, n (%) [95% CI] | 64 (14.7) [12.4–17.1] | 60 (13.8) [11.5–16.1] |
| Length for age Z-score, mean (SD)[b] | -0.79 (1.6) | -0.68 (3.6) |
| Underweight, n (%) [95% CI] | 56 (12.9) [10.7–15.4] | 46 (10.6) [8.5–12.6] |
| Weight for age Z-score mean (SD)[b] | -0.87 (1.2) | -0.75 (3.5) |
| ***Asymptomatic malaria prevalence, n (%)*** | 216 (24.2%) | 224 (24.6%) |
| **Bednet used the previous night, n (%)** | | |
| Yes | 655 (90.7) | 681 (93.0) |
| No | 67 (9.3) | 52 (7.0) |
| **Malaria parasitaemia, n, geometric mean, 95% CI count/μL** | 209, 2713.0 (2197.9–3349.0) | 218, 2806.6 (2236.4–3522.3) |
| **Household head education, n (%)** | | |
| None | 241 (33.3) | 228 (31.1) |
| Basic education | 442 (61.1) | 439 (59.8) |
| Secondary education and above | 40 (5.6) | 67 (9.1) |
| **Moderate anaemia prevalence, n (%)** | 325 (36.4) | 320 (35.1) |
| **Socioeconomic status of household heads, n (%)** | 855 (100) | 890 (100) |
| High | 250 (29.2) | 297 (33.4) |
| Low | 605 (70.8) | 593(66.6) |

Frequency (n) and percentages (%) of participants, age in months (mo), standard deviation (SD), 95% confidence interval (95% CI).

[a] Unit of analysis represents individual participant unless if not stated.

[b] Applying approximated WHO growth reference charts (±2 SD).

## Iron-MNP supplementation decreased anaemia in the study children

The prevalence of anaemia was similar between the iron and non-iron group at baseline (36.84% versus 35.66%, p = 0.618) (Fig 2). Overall, endline prevalence of anaemia was 58.6% (N = 1059, 95% CI: 56.3% - 60.9%) and was more prevalent in children in the non-iron group than in those who received the iron intervention (62.63% versus. 55.52%, p = 0.003) (Table 2). Endline prevalence of moderate and severe anaemia were 52.7% (N = 951, 95% CI: 50.3%– 55.0%) and 6.0% (N = 108, 95% CI: 5.0% - 7.2%) respectively (Table 2). Additionally, the risk adjusted logistic regression analysis also indicated that the odds of anaemia was significantly higher in the children from the non-iron group compared to those in the iron group (Table 3). In comparing the difference (endline versus baseline) in anaemia prevalence for the iron and non-iron group, anaemia prevalence was 8.29% lower in the children who received the iron supplementation (Fig 2 and Table 4).

## Iron-MNP supplementation decreased iron deficiency in the children

Baseline adjusted mean ZPP concentration among children in the non-iron group was similar compared to the iron group (121·8 for non-Fe versus 121·3 µmol / mol of haeme for iron children) (p = 0·979) (Table 2). Similarly, endline adjusted mean ZPP concentration was comparable between the non-iron and the iron group (205·1 for non-iron versus 173·6 µmol / mol of haeme for iron children) (p = 0·101) (Table 2). The prevalence of endline iron deficiency was 24.5% (N = 443, 95% CI: 22.6% - 26.6%) (Table 2). The risk-adjusted logistic regression also

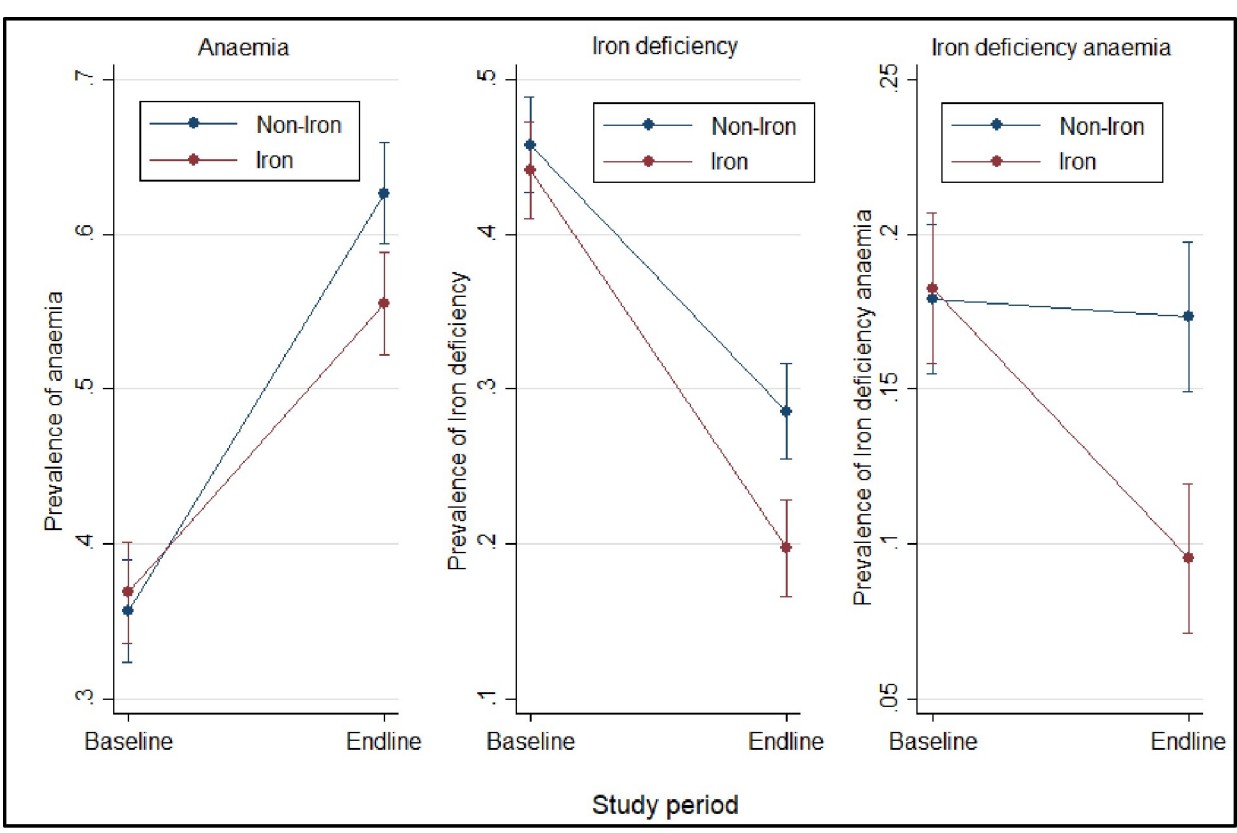

**Fig 2. Prevalence of anaemia, iron status between groups at baseline and endline.** Plot shows anaemia and iron status prevalence with 95% confidence intervals for each group.

**Table 2. Impact of MNP on anaemia and iron status among study participants.**

| Characteristics | Baseline | | | Endline | | |
|---|---|---|---|---|---|---|
| | Iron Group (n = 894)[a] | Non-iron Group (n = 912)[a] | p-values | Iron Group (n = 894)[a] | Non-iron Group (n = 912)[a] | p-values |
| **Anaemic status** | 894 (100) | 912 (100) | | 894 (100) | 912 (100) | |
| Non anaemia, n (%) | 568 (63.5) | 592 (64.9) | | 401 (44.9) | 346 (37.9) | |
| Moderate anaemia, n (%) | 326 (36.5) | 321 (35.1) | 0.56 | 455 (50.9) | 496 (54.4) | 0.03 |
| Severe anaemia, n (%) | - | - | | 38 (4.2) | 70 (7.7) | |
| **Iron status** | | | | | | |
| Iron deficiency, n (%) | 396 (44.3) | 422 (46.3) | 0.41 | 178 (19.9) | 265 (29.1) | < 0.0001 |
| Iron deficiency anaemia, n (%) | 163 (18.2) | 163 (17.9) | 0.84 | 85 (9.5) | 158 (17.3) | < 0.0001 |

[a] Unit of analysis represents individual participant unless if not stated, standard deviation (SD), number (n), percentage (%), 95% confidence interval (95% CI), Two-sample Wilcoxon rank-sum (Mann-Whitney) for continuous variables and Chi squared/ Fisher's exact tests for categorical variables.

indicated that the chances of iron deficiency in children in the non-iron category were significantly higher than in those in the iron group (Table 3). At baseline, there was no significant difference in the prevalence of iron deficiency between the children in the iron and non-iron arm (44.11% versus. 45.77%) (Table 4). Furthermore, at endline, the prevalence of iron deficiency was significantly lower in children who took the iron supplementation (19.71% versus. 28.56%) (Fig 2 and Table 4).

## Iron-MNP supplementation decreased iron deficiency anaemia in the study children

The iron deficiency anaemia endline prevalence was 13.5% (N = 243, 95%: 12.0% - 15.1%) (Table 2). Moreover, the risk-adjusted logistic regression shows that the odds of iron deficiency anaemia was higher in non-iron group compared to the iron group (Table 3). The prevalence of baseline iron deficiency anaemia was comparable between the iron and non-iron participants (Table 4). At endline, the iron group had a substantially lower prevalence of iron deficiency anaemia compared to non-iron group (Table 4).

## Discussion

In this study, the use of daily prophylactic micronutrient powder with or without iron as a fortificant in semi-solid meals in children aged 6 to 35 months was assessed for its impact on blood haemoglobin levels, anaemia, iron deficiency (ID) and iron deficiency anaemia (IDA)

**Table 3. Effect of MNP on anaemia and iron status among study children.**

| Outcomes | Study groups | Number (% of children with outcome) | Adjusted odds ratio [g] (95% CI) | p-values |
|---|---|---|---|---|
| **Anaemia** | Non-iron | 566 (62.1%) | 1 | 0.002 |
| | Iron | 493 (55.1%) | 0.72 (0.59–0.89) | |
| **Iron deficiency** | Non-iron | 265 (29.1%) | 1 | < 0.001 |
| | Iron | 178 (19.9%) | 0.59 (0.47–0.75) | |
| **Iron deficiency anaemia** | Non-iron | 158 (17.3%) | 1 | < 0.001 |
| | Iron | 85 (9.5%) | 0.47 (0.35–0.64) | |

95% confidence interval (95% CI), percentage (%), g: Adjusted for child's age and sex. Further adjusted for baseline anaemia, baseline iron deficiency and baseline iron deficiency anaemia in the model for anaemia, iron deficiency and iron deficiency anaemia respectively.

**Table 4. Effect of MNP on haemoglobin, anaemia, iron biomarkers and CRP among study participants.**

| Characteristics | Baseline | | | | Endline | | | | Difference in difference | | |
|---|---|---|---|---|---|---|---|---|---|---|---|
| | Iron Group | Non-iron Group | Difference | p-values | Iron Group | Non-iron Group | Difference | p-values | Difference | 95% CI | p-values |
| **Haemoglobin status** | | | | | | | | | | | |
| Haemoglobin mean (SD) (g/dL) | 10.29 (± 1.26) | 10.29 (± 1.27) | -0.01 | 0.904 | 9.65 (± 1.70) | 9.31 (± 1.52) | 0.34 | <0.001 | 0.35 | 0.19, 0.51 | <0.001 |
| **Anaemia status** | | | | | | | | | | | |
| Anaemia, (%) | 36.84 | 35.66 | 1.18 | 0.618 | 55.52 | 62.63 | -7.11 | 0.003 | -8.29 | -13.96, -2.63 | 0.004 |
| **Iron biomarkers status** | | | | | | | | | | | |
| Ferritin (µg/L) | 86.99 | 72.05 | 14.94 | 0.263 | 173.24 | 147.80 | 25.44 | 0.056 | 10.50 | -25.61, 46.61 | 0.569 |
| Ferritin adjusted (µg/L) | 62.00 | 61.63 | 0.37 | 0.975 | 142.73 | 115.67 | 27.06 | 0.026 | 26.69 | -5.87, 59.24 | 0.108 |
| Transferrin (ng/m/L) | 81.08 | 86.23 | -5.15 | 0.324 | 63.16 | 78.29 | -15.13 | 0.003 | -9.97 | -23.69, 3.74 | 0.154 |
| Zinc protoporphyrin (µmol ZPP/mol haeme) | 123.91 | 125.52 | -1.61 | 0.918 | 179.29 | 213.85 | -34.56 | 0.027 | -32.95 | -74.08, 8.18 | 0.116 |
| Zinc protoporphyrin adjusted (µmol ZPP/mol haeme) | 121.30 | 121.79 | -0.49 | 0.979 | 173.59 | 205.13 | - 31.54 | 0.101 | -31.05 | -81.39, 19.28 | 0.227 |
| Iron deficiency (%) | 44.11 | 45.77 | -1.66 | 0.457 | 19.71 | 28.56 | -8.85 | <0.001 | - 7.19 | -12.90, -1.49 | 0.013 |
| Iron deficiency anaemia (%) | 18.25 | 17.90 | 0.35 | 0.839 | 9.53 | 17.35 | -7.82 | <0.001 | -8.18 | -12.63, -3.72 | <0.001 |
| **Inflammation** | | | | | | | | | | | |
| C-reactive protein (mg/L) | 3.37 | 3.26 | 0.11 | 0.708 | 4.06 | 3.97 | 0.09 | 0.729 | -0.01 | -0.72, 0.70 | 0.983 |

In the adjusted ZPP, participants with abnormally high CRP levels (due to infection) were excluded. 95% confidence interval (95% CI), percentage (%).

status. This was a secondary analysis of a previous double-blind, cluster randomized trial which assessed the effect of iron supplementation on malaria incidence among pre-school children in rural Ghana [25]. Iron deficiency and iron deficiency anaemia IDA prevalence improved at the end of the intervention among the children in the iron group than those in the non-iron group. These results concord with other iron-MNP studies [34,35]. None the less, other studies have linked ID and IDA prevalence to the impact of MNP intervention equivocally [36–38]. The MNP used in the current study also contained ascorbic acid which is thought to stimulate iron absorption into mucosal cells by preventing the formation of insoluble and unabsorbable iron compounds [39,40] Of the other components of the MNP, the effect of vitamin A on iron nutrition remains unclear. Some studies found low doses of vitamin A like what was used in the current study to enhance iron absorption [41,42] although high doses (such as 1800 µg) may be detrimental to iron nutrition [43]. In other studies vitamin A did not enhance iron absorption in humans [44]. However, simultaneous use of iron and vitamin A supplements appears to mitigate iron deficiency anaemia [45].

In endemic areas, malaria is a major contributor to anaemia due to haemolysis of both parasitized and unparasitized erythrocytes by the immune system and is characterised by temporary bone marrow dyserythropoiesis [24,46]. In our study, there was no difference in asymptomatic malaria prevalence and parasitaemia between children in the iron-MNP and the non-iron MNP groups at baseline. Indeed, the incidence of malaria was not different between the groups at the end of the study as previously reported [25]. A previous sub-group

analysis of baseline haemoglobin and ZPP concentrations indicated that children from the iron group who were iron replete (ZPP $\leq$ 52 μmol / mol of haeme) with moderate anaemia had reduced risk of uncomplicated malaria or severe malaria compared to non-iron group children who were iron replete and moderately anaemic at baseline [25]. Paradoxically, whole blood ZPP levels were limited in distinguishing between those with and without an iron deficiency. However, no additional predictive benefit was found even when paired with levels of haemoglobin, but rather overestimated ID prevalence relative to standard cut-off points (> 52 μmol / mol haeme). These findings from our study is consistent with other MNP studies that used ZPP as additional iron indicator [31,47,48], but contrary to other MNP intervention findings [31,49]. This may be attributed to the differences in ZPP cut-offs used to define iron deficiency, which was lower in our study (> 52 μmol / mol of haeme) compared to a study in Zanzibar (> 80 μmol / mol of haeme) [31,49,50]. During intraerythrocytic malaria parasiteamia, most of the host haemoglobin haeme is converted into a nontoxic haeme crystal, haemozoin [51]. Though, erythrocyte zinc protoporphyrin IX is normally present at 0.5 μM, with a ratio of 1:40000 haemes, it can rise 10-fold in certain anaemias and iron deficients or depletes which is associated with high CRP levels and malaria immunity [52]. Several ZPP cut-offs for defining iron deficiency have been proposed based on the population being studied and the specimen processing method (e.g. washed versus unwashed red blood cells). Blood samples in our study were washed prior to analysis. It appears unwashed red blood cells was used in the Zanzibar trial and this may have led to the higher ZPP cut-off used [19]. The differences in iron status classification between these studies may have affected the comparability of statistical outcomes. One main common finding between our study and that performed in Zanzibar was the significant protective effect of iron to mitigate the risk of iron deficiency and anaemia among young children.

In 2009, a systematic review of 14 studies reported that the provision of iron mitigated the risk of malaria but the effect was reversed when routine malaria management and surveillance were absent [53]. However, none of the studies included iron fortification intervention trials. Our study used powdered iron fortificant (ferrous fumarate) with different absorption characteristics from the iron supplement (provided in the form of iron and folic acid tablets) used in other studies [19]. In addition, iron (ferrous fumarate) microencapsulation may have preserved iron in the food matrix from oxidation and likely delayed peak plasma iron concentrations. [54–56]. This may have reduced the level of freely accessible iron in circulation and mitigated the risk of malaria in the study children [25,57]. In our study, majority of children slept under bed net and at the end of the intervention, there was no increase in malaria incidence in the iron group despite the substantial nutritional benefit observed. Findings from our study support other studies underscoring the importance of iron-MNP as an intervention measure to minimize iron deficiency and associated consequences in malaria endemic areas when adequate malaria infection prevention measures are in place.

## Conclusion

Prophylactic iron-MNP continued use decreased anaemia, iron deficiency and iron deficiency anaemia among pre-school children living in rural Ghana's malaria endemic area.

## Acknowledgments

We would like to thank the study participants and their caregivers; the KHRC field team and staff; chiefs, opinion leaders and elders of participating communities; participating health facilities; the GHS staff in Wenchi and Tain; the Ethics Boards of KHRC, GHS and SickKids of Canada; the DSMB; and FDA of Ghana.

## Author Contributions

**Conceptualization:** Samuel Kofi Tchum, Fareed Kow Arthur, Bright Adu, Samuel Asamoah Sakyi, Latifatu Alhassan Abubakar, Dorcas Atibilla, Felix Boakye Oppong, Francis Dzabeng, Benjamin Amoani, Thomas Gyan, Emmanuel Arhin, Kwaku Poku-Asante.

**Data curation:** Samuel Kofi Tchum, Fareed Kow Arthur, Samuel Asamoah Sakyi, Seeba Amenga-Etego, Felix Boakye Oppong, Francis Dzabeng, Benjamin Amoani, Thomas Gyan, Kwaku Poku-Asante.

**Formal analysis:** Samuel Kofi Tchum, Bright Adu, Samuel Asamoah Sakyi, Dorcas Atibilla, Felix Boakye Oppong, Francis Dzabeng, Benjamin Amoani, Kwaku Poku-Asante.

**Funding acquisition:** Samuel Kofi Tchum, Kwaku Poku-Asante.

**Investigation:** Samuel Kofi Tchum, Fareed Kow Arthur, Bright Adu, Samuel Asamoah Sakyi, Latifatu Alhassan Abubakar, Dorcas Atibilla, Kwaku Poku-Asante.

**Methodology:** Samuel Kofi Tchum, Fareed Kow Arthur, Bright Adu, Samuel Asamoah Sakyi, Latifatu Alhassan Abubakar, Dorcas Atibilla, Felix Boakye Oppong, Francis Dzabeng, Benjamin Amoani, Thomas Gyan, Emmanuel Arhin, Kwaku Poku-Asante.

**Project administration:** Samuel Kofi Tchum, Latifatu Alhassan Abubakar, Dorcas Atibilla, Kwaku Poku-Asante.

**Resources:** Samuel Kofi Tchum, Benjamin Amoani, Kwaku Poku-Asante.

**Software:** Samuel Kofi Tchum, Seeba Amenga-Etego, Felix Boakye Oppong, Francis Dzabeng, Kwaku Poku-Asante.

**Supervision:** Samuel Kofi Tchum, Fareed Kow Arthur, Bright Adu, Samuel Asamoah Sakyi, Emmanuel Arhin, Kwaku Poku-Asante.

**Validation:** Samuel Kofi Tchum, Fareed Kow Arthur, Bright Adu, Samuel Asamoah Sakyi, Latifatu Alhassan Abubakar, Dorcas Atibilla, Seeba Amenga-Etego, Felix Boakye Oppong, Francis Dzabeng, Benjamin Amoani, Thomas Gyan, Kwaku Poku-Asante.

**Visualization:** Fareed Kow Arthur, Seeba Amenga-Etego, Felix Boakye Oppong, Thomas Gyan, Kwaku Poku-Asante.

**Writing – original draft:** Samuel Kofi Tchum, Fareed Kow Arthur, Bright Adu, Samuel Asamoah Sakyi, Kwaku Poku-Asante.

**Writing – review & editing:** Samuel Kofi Tchum, Fareed Kow Arthur, Bright Adu, Samuel Asamoah Sakyi, Latifatu Alhassan Abubakar, Dorcas Atibilla, Seeba Amenga-Etego, Felix Boakye Oppong, Francis Dzabeng, Benjamin Amoani, Thomas Gyan, Emmanuel Arhin, Kwaku Poku-Asante.

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
