## [Decision Letter · Decision Letter 0]

18 Dec 2020

PONE-D-20-35871

Impact of Iron Fortification on Anaemia and Iron Deficiency among Pre-school Children Living in Rural Ghana

PLOS ONE

Dear Dr. Tchum,

Thank you for submitting your manuscript to PLOS ONE. After careful consideration, we feel that it has merit, and indeed both reviewers are positive regarding your work. However, this latter does not fully meet PLOS ONE’s publication criteria as it currently stands. Therefore, we invite you to submit a revised version of the manuscript that addresses the points raised during the review process.

Notably, you will see that both reviewers asked for additional information, and one reviewer made comment regarding the statistical analysis performed. In addition, one reviewer indicated some mistakes in the manuscript that need to be taken into consideration. Please double check for this.

We look forward to receiving your revised manuscript.

Kind regards,

Francois Blachier, PhD

Academic Editor

PLOS ONE

2. Please upload a copy of Figure 2, to which you refer in your text on page 12. If the figure is no longer to be included as part of the submission please remove all reference to it within the text.

Reviewers' comments:

Reviewer's Responses to Questions

**Comments to the Author**

1. Is the manuscript technically sound, and do the data support the conclusions?

Reviewer #1: Yes

Reviewer #2: Yes

2. Has the statistical analysis been performed appropriately and rigorously? 

Reviewer #1: Yes

Reviewer #2: Yes

3. Have the authors made all data underlying the findings in their manuscript fully available?

Reviewer #1: Yes

Reviewer #2: Yes

4. Is the manuscript presented in an intelligible fashion and written in standard English?

Reviewer #1: Yes

Reviewer #2: Yes

5. Review Comments to the Author

Reviewer #1: My compliments to the authors, this is a well written manuscript about a properly designed and explained study.

I have a few questions/remarks though:

- The authors mention the definitions of anaemia, ID etc in the statistics paragraph of the Methods section. I suggest to make a separate paragraph. Also, please add references regarding the definitions and cut-off levels. Have the authors used WHO guidelines? I believe not in the case of ZPP. And what is adjusted ZPP (Table 4?) Can they also elaborate on the acute phase aspect of ZPP?

- In the Results section and baseline characteristics table, it is technically wrong to perform statistical analyses on the baseline characteristics of a randomized trial. Any unsimilar data have to be coincidences if randomization was properly done. Suggest to remove the p-values and just note that both groups are comparable.

- Was the cluster randomization performed with a computer? Please elaborate.

- The authors mention in the Discussion the hypothesis that malaria affects hepcidin and hereby iron homeostasis bij leading to upregulated hepcidin with followed lower iron status. However, the authors do not have hepcidin levels right? If not, this information is not very relevant for this article.

- I miss discussion on the impact of the multi fortification instead of just plane iron supplementation, for example, vit C also stimulates iron absorption. Please elaborate a bit.

Reviewer #2: The manuscript PONE-D-20-35871 titled “Impact of iron fortification on anemia and iron deficiency among pre-school children living in rural Ghana” assesses in a randomized fashion the impact of long-term prophylactic iron (ferrous fumarate), administered as a micronutrient powder along with vitamin A, ascorbic acid and zinc, on the risk of iron deficiency (ID) and iron deficiency anemia (IDA) in Ghanaian infants and toddlers aged 6 to 35 months living in a rural malaria endemic area. The results clearly show that the use of the selected iron-fortified MNP reduced anemia, ID and IDA in this setting compared to a similar MNP without iron. The manuscript is well-written with few minor grammatical mistakes.

Specific comments:

1. A flow diagram would be useful, with further explanation of the reasons for exclusion of 262 children.

2. Figure 1 does not contribute much and should be removed.

3. There is a serious mistake in Table 3. The number (percentage of children with outcome, i.e., anemia, ID and IDA) has been mixed up for the two study groups, i.e., the results of the iron group are reported in the non-iron group and vice versa. Please, correct.

4. The phrase acute phase status on the title of Table 4 is confusing. I would change the title of Table 4 to: Effect of MNP on hemoglobin, anemia, iron biomarkers and CRP among study participants. Moreover, Table 4 has several minor mistakes. The difference of transferrin in baseline between the two study groups (iron versus non-iron) is -5.15 instead of -5.16. In the same column, the difference in baseline IDA between the two groups is 0.35 (not 0.36) and the difference in CRP, also in the same column, is 0.11 instead of 0.10.

5. The authors should stick throughout the manuscript with either the British or the American way of English writing. If haemoglobin and haemolysis are used, then haeme should also be used instead of heme.

6. I believe that the reference on line 106, page 5 should be reference No. 25 and not No. 26.

7. In line 58, page 3, please remove the word of. As an of intervention… should be as an intervention.

8. In line 66, page 3 uninfected red erythrocytes, should be changed to uninfected erythrocytes. The authors can write red cells or erythrocytes but certainly not red erythrocytes.

9. In line 124, page 6, the sentence starting from Sachets containing without or with should be corrected to Sachets without or with iron. Please, remove the word containing.

10. The first sentence of page 9 (line 186), should read: The characteristics of the study population have previously been described elsewhere.

6. PLOS authors have the option to publish the peer review history of their article (what does this mean?). If published, this will include your full peer review and any attached files.

Reviewer #1: No

Reviewer #2: **Yes: **Elpis Mantadakis, MD, PhD

---

## [Author Response · Author response to Decision Letter 0]

8 Jan 2021

Reply to reviewers' comments

Reviewer #1: 

My compliments to the authors, this is a well written manuscript about a properly designed and explained study.

I have a few questions/remarks though:

- The authors mention the definitions of anaemia, ID etc in the statistics paragraph of the Methods section.

1. I suggest to make a separate paragraph

Response: A separate paragraph has been created, page 8 and lines 173-179.

2. Also, please add references regarding the definitions and cut-off levels. Have the authors used WHO guidelines? I believe not in the case of ZPP.

Response: References added in accordance with WHO guidelines including ZPP, page 8 and line 178.

3. What is adjusted ZPP (Table 4?) 

Response: In the adjusted ZPP, participants with abnormally high CRP levels (due to infection) were excluded. Page 15 and lines 289-291.

Can they also elaborate on the acute phase aspect of ZPP? 

Response: During intraerythrocytic malaria parasiteamia, most of the host haemoglobin haeme is converted into a nontoxic haeme crystal, haemozoin. Though, erythrocyte zinc protoporphyrin IX is normally present at 0.5 µM, with a ratio of 1:40000 haemes, it can rise 10-fold in certain anaemias and iron deficients or depletes which is associated with high CRP levels and malaria immunity. Page 19 and lines 346-350.

4. In the Results section and baseline characteristics table, it is technically wrong to perform statistical analyses on the baseline characteristics of a randomized trial. Any unsimilar data have to be coincidences if randomization was properly done. Suggest to remove the p-values and just note that both groups are comparable. 

Response: This has been corrected, page 10 -11 and lines 222-223.

5. Was the cluster randomization performed with a computer? 

Response: Yes, randomization was done with a computer. 

6. Please elaborate.

Response: A cluster comprised one or more households in the same compound with at least one child enrolled and to prevent cross contamination between the groups via food sharing, a cluster randomization design was employed using a computer-generated model. This simple random allocation was carried out at the cluster level because the participants lived in same compound with different or extended families in the rural communities. Therefore, this randomization scheme also facilitated the monitoring of participants, particularly those who moved within or outside the study area. The cluster randomization design or model of the study population have previously been described elsewhere (25). Page 6 and lines 121-130.

7. The authors mention in the Discussion the hypothesis that malaria affects hepcidin and hereby iron homeostasis bij leading to upregulated hepcidin with followed lower iron status. However, the authors do not have hepcidin levels right? 

Response: Yes, the reviewer is right. 

8. If not, this information is not very relevant for this article. 

Response: Information removed accordingly. Page 17 and lines 302-312.

9. I miss discussion on the impact of the multi fortification instead of just plane iron supplementation, for example, vit C also stimulates iron absorption. Please elaborate a bit.

Response: The MNP used in the current study also contained ascorbic acid which is thought to stimulate iron absorption into mucosal cells by preventing the formation of insoluble and unabsorbable iron compounds (1, 2). Page 17 and lines 312-323.

Reviewer #2: 

The manuscript PONE-D-20-35871 titled “Impact of iron fortification on anemia and iron deficiency among pre-school children living in rural Ghana” assesses in a randomized fashion the impact of long-term prophylactic iron (ferrous fumarate), administered as a micronutrient powder along with vitamin A, ascorbic acid and zinc, on the risk of iron deficiency (ID) and iron deficiency anemia (IDA) in Ghanaian infants and toddlers aged 6 to 35 months living in a rural malaria endemic area. The results clearly show that the use of the selected iron-fortified MNP reduced anemia, ID and IDA in this setting compared to a similar MNP without iron. The manuscript is well-written with few minor grammatical mistakes.

Specific comments:

1. A flow diagram would be useful, with further explanation of the reasons for exclusion of 262 children. 

Response: Created a flow diagram with additional explanations for exclusions of 262 children as Fig 1, page 10 and lines 208-212.

2. Supplementary Figure 1 does not contribute much and should be removed.

 Response: Supplementary Fig 1 file removed and replaced with Fig 1 file. 

3. There is a serious mistake in Table 3. The number (percentage of children with outcome, i.e., anemia, ID and IDA) has been mixed up for the two study groups, i.e., the results of the iron group are reported in the non-iron group and vice versa. Please, correct.

Response: Thank you. This has been corrected. Page 15 and lines 285-286.

4. The phrase acute phase status on the title of Table 4 is confusing. I would change the title of Table 4 to: Effect of MNP on hemoglobin, anemia, iron biomarkers and CRP among study participants. 

Response: The title of Table 4 has been changed as suggested by the reviewer. 

Page 16 and lines 288-290. 

5. Moreover, Table 4 has several minor mistakes. The difference of transferrin in baseline between the two study groups (iron versus non-iron) is -5.15 instead of -5.16.

 Response: This has been corrected. Page 16 and lines 290-291. 

6. In the same column, the difference in baseline IDA between the two groups is 0.35 (not 0.36). 

Response: This has been corrected. Page 16 and lines 290-291.

7. The difference in CRP, also in the same column, is 0.11 instead of 0.10.

 Response: This has been corrected. Page 16 and lines 290-291. 

8. The authors should stick throughout the manuscript with either the British or the American way of English writing. If haemoglobin and haemolysis are used, then haeme should also be used instead of heme.

 Response: This has been corrected. British English has been used throughout the manuscript. 

9. I believe that the reference on line 106, page 5 should be reference No. 25 and not No. 26.

 Response: This has been corrected. Page 5 and lines 105-106. 

10. In line 58, page 3, please remove the word of. As an of intervention… should be as an intervention.

 Response: The word ‘of’ has been removed. Page 3 and line 58. 

11. In line 66, page 3 uninfected red erythrocytes, should be changed to uninfected erythrocytes. The authors can write red cells or erythrocytes but certainly not red erythrocytes. 

Response: This has been corrected. Page 3 and line 66.

12. In line 124, page 6, the sentence starting from Sachets containing without or with should be corrected to Sachets without or with iron. Please, remove the word containing.

 Response: The word ‘containing’ has been removed. Page 6 and line 131. 

13. The first sentence of page 9 (line 186), should read: The characteristics of the study population have previously been described elsewhere.

 Response: Thank you. This has been corrected. Page 9 and line 196-197.

---

## [Decision Letter · Decision Letter 1]

19 Jan 2021

Impact of Iron Fortification on Anaemia and Iron Deficiency among Pre-school Children Living in Rural Ghana

PONE-D-20-35871R1

Dear Dr. Tchum,

We’re pleased to inform you that your manuscript has been judged scientifically suitable for publication and will be formally accepted for publication once it meets all outstanding technical requirements.

Kind regards,

Francois Blachier, PhD

Academic Editor

PLOS ONE

Additional Editor Comments (optional):

Reviewers' comments:

Reviewer's Responses to Questions

**Comments to the Author**

1. If the authors have adequately addressed your comments raised in a previous round of review and you feel that this manuscript is now acceptable for publication, you may indicate that here to bypass the “Comments to the Author” section, enter your conflict of interest statement in the “Confidential to Editor” section, and submit your "Accept" recommendation.

Reviewer #2: All comments have been addressed

2. Is the manuscript technically sound, and do the data support the conclusions?

Reviewer #2: Yes

3. Has the statistical analysis been performed appropriately and rigorously? 

Reviewer #2: Yes

4. Have the authors made all data underlying the findings in their manuscript fully available?

Reviewer #2: Yes

5. Is the manuscript presented in an intelligible fashion and written in standard English?

Reviewer #2: Yes

6. Review Comments to the Author

Reviewer #2: The revised manuscript is substantially better than the original submission. The authors addressed all my comments adequately.

7. PLOS authors have the option to publish the peer review history of their article (what does this mean?). If published, this will include your full peer review and any attached files.

Reviewer #2: **Yes: **Elpis Mantadakis, MD, PhD

---

## [Editor Report · Acceptance letter]

28 Jan 2021

PONE-D-20-35871R1 

Impact of Iron Fortification on Anaemia and Iron Deficiency among Pre-school Children Living in Rural Ghana 

Dear Dr. Tchum:

I'm pleased to inform you that your manuscript has been deemed suitable for publication in PLOS ONE. Congratulations! Your manuscript is now with our production department. 

Kind regards, 

on behalf of

Dr. Francois Blachier 

Academic Editor

PLOS ONE